# Evaluation of the *Conversations about Non-Suicidal Self-Injury* Mental Health First Aid Course: Effects on Knowledge, Stigmatising Attitudes, Confidence and Helping Behaviour

**DOI:** 10.3390/ijerph20043749

**Published:** 2023-02-20

**Authors:** Kathy S. Bond, Julia N. Lyons, Fairlie A. Cottrill, Amanda V. Sabo, Simone E. Baillie, Alyssia Rossetto, Louise Kelly, Claire M. Kelly, Nicola J. Reavley, Anthony F. Jorm, Amy J. Morgan

**Affiliations:** 1Mental Health First Aid Australia, Level 18, 150 Lonsdale Street, Melbourne, VIC 3000, Australia; 2Centre for Mental Health, Melbourne School of Population and Global Health, The University of Melbourne, Carlton, VIC 3010, Australia

**Keywords:** non-suicidal self-injury, mental health first aid, education, early intervention, knowledge, confidence, stigma, helping behaviour

## Abstract

Background: Non-suicidal self-injury (NSSI) is a common mental health problem, with a 19% lifetime prevalence in Australian adolescents and 12% in adults. Though rates of professional help-seeking for NSSI are low, disclosure to family and friends is more common, providing opportunities for them to encourage professional support. Mental Health First Aid^®^ Australia’s *Conversations about Non-Suicidal Self-Injury* course provides evidence-based training for the general public to support a person engaging in NSSI. Methods: This uncontrolled trial evaluated the effects of the *Conversations about Non-Suicidal Self-Injury* course on participants’ knowledge, confidence, stigmatising attitudes, and intended and actual helping behaviours. Surveys were administered pre- and post-course, and at a six-month follow-up. A linear mixed-model analysis determined mean change over time, and effect sizes were estimated using Cohen’s d. Course satisfaction was assessed using descriptive statistics and summative content analysis of qualitative data. Results: The pre-course survey was completed by 147 Australian participants (77.5% female, mean age 45.8 years), 137 (93.2%) at post-course and 72 (49%) at follow-up. Knowledge, confidence, quality of intended helping behaviours, and quality of actual helping behaviours increased significantly at both time points. Social distance decreased significantly at all time points and stigma decreased significantly at post-course. The course was perceived to be highly acceptable by participants. Conclusions: There is initial evidence that the *Conversations about Non-Suicidal Self-Injury* course is effective and acceptable for members of the public who may support a person engaging in NSSI.

## 1. Introduction

Non-suicidal self-injury (NSSI) refers to injuries intentionally inflicted upon oneself that are not intended to result in death and are not culturally sanctioned [1,2]. The 2020–2021 Australian National Study of Mental Health and Wellbeing reported a lifetime population prevalence of 8.8% in people aged 16–85 (11.4% in females, 6.2% in males), with substantially higher prevalence in people aged 16–24 (24.7% in females, 12.4% in males) [3]. The lifetime prevalence of NSSI is also higher in clinical populations, with those experiencing a mental illness being 5.5 to 7.7 times more likely to self-injure over the previous 4 weeks than nonclinical populations [4]. For people who have engaged in NSSI at some point in their lifetime, prior experience of a mental illness has been shown to predict the subsequent onset of self-injurious behaviours, with mood disorders showing a particularly strong relationship [5]. It is also common for NSSI to precede the onset and diagnosis of many mental illnesses, suggesting the possibility of a reciprocal relationship between mental illness and NSSI.

The most common reasons for NSSI cited by adolescents and adults are to distract or seek relief from distressing thoughts, feelings, problems, or bad memories [6,7]. Although NSSI is often reported by those who engage in it as an effective coping mechanism to alleviate distress, substantial evidence demonstrates that negative outcomes are associated with this behaviour. Notably, adolescents who engage in NSSI are nearly five times more likely to experience suicidal ideation than those who do not, and those who self-injure more frequently are at substantially higher risk of suicide [7]. Additionally, NSSI in adolescence is associated with an increased risk of depression, anxiety, and substance misuse in later life [8].

Help-seeking rates in people who engage in NSSI are low, with less than 50% seeking emotional support for this behaviour, and only 16% report seeking medical treatment [4]. Voluntary disclosure of NSSI to another person can precipitate professional help-seeking, especially when the disclosure recipient is a peer [9]. Considering adolescents and adults are most likely to disclose their NSSI to a family member or friend [9,10], there are unique opportunities for the people receiving such disclosures to offer support and encourage professional help. In a systematic review of NSSI disclosure responses, Park et al. [11] reported that understated acceptance (as opposed to strong emotional reactions) and ongoing support after the disclosure were perceived as helpful by individuals engaging in NSSI. Similarly, Wadman et al. [12] interviewed adolescents who had disclosed their NSSI to others and found that non-judgemental support from friends helped to prevent future repetition of self-injury behaviours.

Conversely, responses to NSSI involving dismissive, trivialising, or stigmatising attitudes were associated with withdrawal from further help-seeking, increased frequency of NSSI, and increased suicide attempts [11]. As NSSI is a highly stigmatised behaviour [13], feelings of shame, fear of judgement, and not wanting to be a burden on others are further barriers to disclosure and help-seeking [10]. The proximity of family members and friends to those engaging in NSSI means that they are well placed to identify warning signs and respond non-judgementally, which may ameliorate these barriers.

Although family and friends are most likely to notice signs of NSSI and receive NSSI disclosures [10], they report having insufficient knowledge regarding how to help someone engaging in NSSI [14,15]. Mental Health First Aid^®^ Australia develops evidence-based training programs to build community capacity for providing early-intervention mental health support (for more information on Mental Health First Aid training programs and research, see https://mhfainternational.org/ [accessed on 20 February 2023]). Existing Mental Health First Aid (MHFA™) courses cover a range of mental health problems and crises, such as depression, anxiety, and suicidality, with specific guidance for various demographic groups, including for young people, older persons, and cultural groups. In 2018, a four-hour course entitled *Conversations about Non-Suicidal Self-Injury* was developed to train members of the general public (known as first aiders) to identify and appropriately support a person who is engaging in NSSI. The course curriculum was based on a set of guidelines [16] developed using the Delphi expert consensus method, where Australian and international lived experience experts (lived experience experts are people with a personal experience of mental illness or people who care for someone with a personal experience of mental illness) and mental health professionals endorsed statements about what a first aider should know and do when providing support to someone engaging in NSSI. The guidelines were first developed in 2008 [17] and updated in 2014 [18].

MHFA courses have consistently been shown to increase the mental health literacy of first aiders, reduce stigmatising attitudes towards mental health problems, and improve first aiders’ confidence in responding to mental health problems and crises [19,20,21]. It was, therefore, expected that the *Conversations about Non-Suicidal Self-Injury* course would similarly improve participants’ knowledge and skills in providing safe, effective, and appropriate support to people engaging in NSSI. Accordingly, the present study aimed to evaluate the safety, acceptability, and impact of the *Conversations about Non-Suicidal Self-Injury* course on participants’ knowledge about mental health first aid and NSSI, confidence when providing support, stigmatising views about NSSI, quality of intended and actual helping behaviours, and course satisfaction. This evaluation also sought information about possible ways to improve the course materials and teaching outcomes for future iterations of the course curriculum.

## 2. Materials and Methods

### 2.1. Intervention

The *Conversations about Non-Suicidal Self-Injury* course [22] was launched by Mental Health First Aid^®^ Australia in 2018. It is delivered by a licensed Mental Health First Aid^®^ instructor and teaches participants the skills and knowledge needed to support a person engaging in NSSI. Before becoming a *Conversations about Non-Suicidal Self-Injury* instructor, they must first attend an MHFA course themselves, be licensed and have delivered at least two Standard MHFA courses, and have been through the upskilling process to become a *Conversations about Non-Suicidal Self-Injury* instructor. The learning objectives are to understand:Why people engage in NSSI;How to talk to someone about their NSSI;How to help the person stay safe;How to connect someone to appropriate professional help;How to assess for suicidal thoughts and behaviours.

### 2.2. Procedures

A member of the research team contacted instructors who were delivering the *Conversations about Non-Suicidal Self-Injury* course in Australian capital cities between 2018 and 2021 to seek permission to collect data before and after their courses.

Course attendees were invited to complete three surveys: one before the course (pre-course; completed on paper or online via the survey software Survey Monkey [23]), one immediately after the course (post-course; completed on paper), and one six months after the course (follow-up; completed online via Survey Monkey [23]). If a participant did not complete the six-month follow-up survey, the research team followed up with three email reminders and made one telephone call. Appendix A contains copies of the surveys from each time point. Participants were asked to sign a consent form and were given a Plain Language Statement detailing their involvement in the study. The design of this evaluation is similar to the evaluation of the *Conversations about Gambling* course [20].

### 2.3. Measures

In the pre-course survey, participants were asked to provide demographic information, outline any personal and professional experience with NSSI, indicate whether they had participated in any previous training related to mental health or NSSI, and the reason for attending the course. These questions were adapted from a previous evaluation on the *Conversations about Gambling* course [20].

#### 2.3.1. Knowledge about NSSI

At each time point, participants were presented with 16 true or false questions about NSSI that were based on the course content. They were asked to respond with “Disagree”, “Agree”, or “Don’t know” (the latter was coded as an incorrect response). Knowledge scores were calculated as the percentage of correct answers (possible range: 0–100), and mastery of knowledge was set at 80%, consistent with previous evaluations [20]. As this was a criterion-referenced measure, Subkoviak [24] was used to estimate the agreement coefficient of 0.63.

#### 2.3.2. Stigmatising Attitudes and Social Distance

At each time point, participants were presented with a vignette that depicted Alicia, an 18-year-old girl who is described as “your niece”, showing warning signs for NSSI, such as “bruising and scratches on her arms and legs” and “wearing her winter clothes in spite of the heat” (see Appendix A). Consistent with previous course evaluations [20], the vignette was developed using the proposed DSM-5 criteria for NSSI disorder [25]. The same vignette was used at each time point to minimise potential confounders (e.g., discrepancies in interpreting different vignettes) and to reliably assess change over time. At each time point, participants were presented with 7 statements that measured stigma towards Alicia and 7 statements that measured social distance towards “people with a problem like Alicia’s.” Participants indicated how strongly they agreed or disagreed with each statement using a 5-point Likert scale. When the statements are combined into a single value, total mean score produced a possible range between 1 (low) and 5 (high) for stigma and for social distance. The items formed two validated scales based on the work of Griffiths et al. [26], Yap et al. [27], and Link et al. [28]. The stigma scale had an omega value of 0.86 and the social distance scale had a value of 0.92.

#### 2.3.3. Confidence in Intended Helping in Response to the Alicia Vignette

Participants were asked how confident they were in their ability to help Alicia on a 5-point Likert scale from “Not at all confident” (a score of 1) to “Extremely confident” (a score of 5).

#### 2.3.4. Quality of Intended Helping Behaviours in Response to the Alicia Vignette

At each time point, participants were asked to rate how likely they would be to take 18 actions to support Alicia using a 5-point Likert scale from “Very unlikely” (a score of 1) to “Very likely” (a score of 5). Ten actions contrary to the course teachings and underlying guidelines [16] formed a scale of non-recommended helping actions and eight actions consistent with the course and guidelines formed a scale of recommended actions. Quality of intended helping behaviours was assessed by calculating the number of recommended actions that participants rated themselves as “Likely” or “Very likely” to undertake (possible range: 0–8) and the number of non-recommended actions participants rated themselves as “Unlikely” or “Very unlikely” to undertake (possible range: 0–10). A cut-off score for mastery was set at 80% concordance with the course teachings for both scales, consistent with previous evaluations [20]. For recommended actions, participants needed to score at least 7 out of 8 to achieve mastery. For non-recommended actions, participants needed to score at least 8 out of 10 to achieve mastery (i.e., participants reported that they did not intend to perform 8 or more non-recommended actions). Based on a single administration of the measure, the agreement coefficient, determined using Subkoviak [24], was 0.76 for recommended actions and 0.72 for non-recommended actions. There was no significant correlation between recommended and non-recommended actions (*r* = 0.14, 95% confidence interval [CI] −0.02–0.29).

#### 2.3.5. Confidence in and Quality of Actual Helping Behaviours

In the pre-course and six-month follow-up surveys, participants were asked: “In the past 6 months, have you had contact with someone who you thought might be engaging in non-suicidal self-injury?” Those who answered “Yes” were asked how many people they assisted. They were then asked to provide the age range, gender, and type of relationship (e.g., family member or work colleague) of the person they had the most contact with.

Participants were asked to select which, if any, of 20 possible actions they took to support the person they knew. These actions were the same as those assessing intended helping actions, plus two additional items (“I did not do anything” and “I did something else”). Participants who selected “I did something else” were asked to specify what they did in a free-text box. They were then asked how confident they were in their ability to help the person on a 5-point Likert scale from “Not at all confident” (a score of 1) to “Extremely confident” (a score of 5).

Participants who reported helping a person engaging in NSSI were asked two open-ended questions with free-text response options: “What were the effects on the person of what you did?” and “What did the person do as a result of your help?” Participants who reported that they did not offer help were asked “Are there any particular reasons that you did not try to help?” and were prompted to specify these reasons in a free-text box.

As with intended helping behaviour, separate scales were created for recommended and non-recommended actual helping behaviours. Concordance was calculated by summing the number of selected actions that were consistent with the guidelines and course curriculum separately for recommended and non-recommended actions. A cut-off score for mastery was set at 80% of concordant actions for both scales. This translates to a score of 7 or more of the 8 recommended actions and 8 or more of the 10 non-recommended actions. The agreement coefficient, determined from the subset of participants that responded to these items, was 0.82 for recommended actions (n = 85) and 0.83 for non-recommended actions (n = 84) [29]. There was no significant correlation between recommended and non-recommended actions (*r* = −0.001, 95% CI −0.22–0.21).

#### 2.3.6. Course Satisfaction

In the post-course survey, participants were asked to provide information about course satisfaction, including how new, understandable, and relevant the information in the course was; how well it was presented; and how much they liked the course materials. A 5-point Likert scale was used for each question. Participants were also asked to specify what aspects of the course they found most helpful and what could be improved using a free-text response.

### 2.4. Statistical Analysis

Data were analysed using linear and logistic mixed models. Mixed models retain all available data and provide an intention-to-treat estimate of change under the assumption of missingness at random. Models included a fixed effect of time and a random effect of participants to adjust for the correlation of responses within participants over time. Logistic regression was used to explore predictors of missingness at six-month follow-up, using participant demographics and pre-course outcomes as potential predictors of attrition. Age was associated with missingness at follow-up (*p* = 0.046) and was included as a fixed effect to help meet the missing at random assumption. Planned comparisons investigated change over time between pre-course and post-course outcomes and between pre-course and six-month follow-up outcomes. Variables with skewed distributions resulting in skewed residuals were transformed. Where this was unsuccessful, bootstrapping and calculation of bias-corrected parameter confidence intervals was performed to test the robustness of the estimates reached using conventional methods.

The stigma variable was highly skewed; therefore, a linear model was deemed inappropriate. For this outcome, mean scores were dichotomised based on scoring 1 (the lowest possible score) or scoring greater than 1 (indicating some stigma). A mixed-effect logistic regression model was used to calculate the odds of scoring 1 on stigma (low stigma) after the course. Effect sizes were calculated and interpreted using Cohen’s d and Cohen’s criteria for small, medium, and large effects, where the difference between means was divided by their pooled standard deviation [30]. Scales were formed from multiple items with respondents who had answered at least 80% on each item in the scale. Estimates were presented with 95% CIs and the significance level was set at *p* < 0.05. Analyses were performed in Stata 17.

Course satisfaction data were analysed using means and standard deviations. Content analysis was used to determine prominent themes in the free-text responses [31], accompanied by the frequency of each theme reported as a percentage of total responses.

## 3. Results

### 3.1. Participant Demographics

Between 2018 and 2021, 153 course attendees from 15 courses were approached to participate in the evaluation and 147 (96.1%) consented to participate. Of the 147 participants, 137 (93.2%) provided at least some data at the post-course survey and 72 (49%) provided at least some data at six-month follow-up. Missingness at follow-up was associated with age of the participant, where, for each additional 10 years of age, the odds of completing the follow-up survey increased 30% (*p* = 0.046; 95% CI 1.00, 1.05). No other predictors of attrition were significant.

Participant characteristics are shown in Table 1. The mean age of participants was 45.8 years; over three quarters were female and just over 1% identified as gender diverse. The most commonly held highest level of education was a university degree (42.9%), followed by a certificate, trade, or apprenticeship (30.6%). One person identified as Aboriginal or Torres Strait Islander (0.7%) and ten people spoke a language other than English at home (6.8%). While 102 participants (69.4%) had previous mental health training, only 19 participants (12.9%) had previous training about NSSI. Participants were asked what experience they had with people who engage in NSSI; over a third reported their experience with clients/patients (35.4%), followed by family (29.4%) and no experience (25.2%). Eleven participants (7.5%) also reported that they had personal experience of NSSI. The majority of participants completed the course as part of continuing education for their workplace/profession (70.6%).

### 3.2. Knowledge about NSSI

Mean knowledge about NSSI at pre-course was moderate (see Table 2). The highest proportion of mastery of knowledge score was achieved at post-course, with 89% of participants scoring at least 80% (13 out of 16 correct responses). This had increased from 36% (n = 53) of participants achieving mastery at pre-course and reduced slightly to 76% at follow-up (n = 50). The proportion of participants meeting mastery improved significantly over time at post-course (*p* < 0.001) and at follow-up (*p* < 0.001) when compared to the pre-course proportion. Effect sizes were large at both post-course with a Cohen’s d of 1.32 (95% CI 1.06, 1.58) and at follow-up 0.97 (95% CI 0.66, 1.27; see Table 3).

### 3.3. Social Distance

Mean social distance scores were low at pre-course (m = 1.90, SD = 0.70; see Table 2) and significantly decreased at both post-course and six-month follow-up when compared to pre-course estimates (*p* < 0.001; see Table 3). Effect sizes from pre-course to post-course (Cohen’s d = −0.33) and pre-course to follow-up (Cohen’s d = −0.41) were medium in size (see Table 3).

### 3.4. Stigma

Stigmatising attitudes at pre-course were low (m = 1.67, SD = 0.55) and decreased at post-course and follow-up (see Table 2). At pre-course, 17% (n = 25) of participants produced the lowest score of 1, with this proportion increasing to 32% (n = 43) at post-course and 24% (n = 16) at follow-up. There was a significant reduction in stigmatising attitudes from pre-course to post-course (odds ratio = 0.30; 95% CI 0.14, 0.63) but this reduction was not significant at six-month follow-up (see Table 3).

### 3.5. Confidence in Intended Helping Behaviours in Response to the Alicia Vignette

Mean confidence at pre-course was moderate (m = 2.89, SD = 0.89) and increased significantly over time (*p* < 0.001) by more than 1 point for each time point comparison. These increases produced a large effect size at both pre-course to post-course (d = 1.41) and pre-course to follow-up (d = 1.19; see Table 3).

### 3.6. Quality of Intended Helping Behaviours in Response to the Alicia Vignette

#### 3.6.1. Recommended Actions

The most frequently endorsed recommended intended action at all time points was “Tell Alicia that there are sources of help and support available”, with over 97% of participants selecting this response. Table 4 shows the proportion of participants endorsing each recommended action to support Alicia.

At pre-course, the mean number of intended recommended actions endorsed was 7.19 out of 8 total actions (SD = 1.26; see Table 2). This increased significantly at subsequent time points. These effect sizes were medium and small at both post-course (d = 0.42) and follow-up, respectively (d = 0.29; see Table 3). The proportion of participants who achieved mastery of intended recommended actions was high at pre-course, with 83.9% of people correctly selecting at least seven out of eight recommended actions. This remained above 90% at both post-course and follow-up (see Table 4). The proportion of participants who achieved mastery significantly improved from pre-course to post-course (*p* = 0.01) but not at follow-up (*p* = 0.05; see Table 4).

#### 3.6.2. Non-Recommended Actions

The non-recommended action “Ask Alicia about why she is injuring herself” was incorrectly endorsed by the majority of participants at each time point (73.4% at pre-course, 88.1% at post-course, 77.6% at follow-up; see Table 5). For the purpose of the present evaluation, this item was removed from the scale and subsequent analysis due to its ambiguity and a lack of clarity in the teaching materials regarding whether it was a recommended or non-recommended action. After this item was removed, the most frequently endorsed non-recommended action was “Let Alicia know how distressing her injuries are to you”, with approximately one third of participants incorrectly endorsing this action at each time point. Table 5 shows the proportion of participants endorsing each non-recommended action to support Alicia.

At pre-course, the mean number of concordant actions (i.e., participants correctly not endorsing actions that are not recommended in the training or guidelines) was 6.89 (SD = 1.92) out of 9 total actions. Participants had a significant mean increase of approximately one action (m = 1.07, 95% CI 0.79, 1.34, *p* < 0.001; see Table 3) at post-course when compared to pre-course, producing a medium effect size (d = 0.60, 95% CI 0.36, 0.84; see Table 3). About half of participants achieved mastery of non-recommended actions at pre-course, and this proportion increased significantly at both post-course (*p* = 0.004) and follow-up (*p* < 0.001; see Table 5).

### 3.7. Confidence in Actual Helping Behaviours

Respondents were moderately confident about the help they provided at pre-course, with a mean value of 2.98 (SD = 0.98), which increased to 3.80 (SD = 0.81) at follow-up (see Table 2). Confidence in helping a person who engaged in NSSI improved significantly over time, with a mean increase of 0.79 (95% CI 0.44, 1.13, *p* < 0.001; see Table 3).

### 3.8. Quality of Actual Helping Behaviours

A total of 85 (57.8%) participants reported having contact with someone who had engaged in NSSI at pre-course and 29 (40.2%) reported having contact with someone at follow-up. Of these, 23 people reported having contact with someone at both pre-course and follow-up. These individuals were most commonly the participant’s client or patient (32.9% at pre-course and 41.4% at follow-up) or a family member (23.5% at pre-course and 34.5% at follow-up).

#### 3.8.1. Recommended Action*s*

Participants were asked what actions they took to help the person they had the most contact with in the last six months. The most common recommended action taken was “Help them find ways to make their life more manageable or reduce their distress” at pre-course and at follow-up (see Table 6). Similarly, the second most common action taken was “Asked them about their feelings that have led them to injure themselves” at both time points.

The mean number of recommended actions taken at pre-course was 5.27 out of 8 (SD = 2.42), which increased to 5.48 at follow-up (SD = 2.21; see Table 2), though this was not a significant increase over time (*p* = 0.752; see Table 3). Mastery of recommended actions taken was achieved by 42.4% of participants at pre-course and by 44.8% of participants at follow-up. The proportion of participants who achieved mastery did not significantly change over time (see Table 6).

#### 3.8.2. Non-Recommended Actions

The most common non-recommended action taken at pre-course was “Ask them about why they were injuring themselves” at 48.8%, followed by “Let them know how distressing their injuries were to you” at 17.9% (see Table 7). A small percentage of participants who reported they had contact with someone who engaged in NSSI said they did not do anything before the course (2.4%), which reduced to 0 at follow-up.

The mean number of responses that were concordant with the guidelines (i.e., participants correctly avoided undertaking actions that were not recommended) was 9.51 (SD = 0.83) and increased to 9.83 (SD = 0.47; see Table 2) and showed a small significant increase over time (*p* = 0.009; see Table 3). Mastery of non-recommended actions was achieved by 90.5% of participants prior to the course and was achieved by 100% of participants at follow-up. The proportion of participants who achieved mastery did not significantly increase over time (see Table 7). Supplementary analyses showed that, among the participants who helped at both timepoints, there was also no significant change in the proportion of participants who achieved mastery.

### 3.9. Effects of First Aid

Table 8 summarises codable responses to the questions “What were the effects on the person of what you did?” and “What did the person do as a result of your help?” A codable element within a response was a coherent word or phrase that directly answered the question posed, i.e., could be interpreted as an effect, action, or reason for not helping.

For the first question, 73 responses were recorded before the course and 26 responses were recorded at six-month follow-up. A total of 11 responses were unclear or did not answer the question, resulting in 88 valid responses across both time points, producing 124 codable effects of the help provided. The most frequently reported effects included a perceived reduction in the person’s distress, improved communication between the person and the first aider, a sense that the person felt supported, and subsequent professional help-seeking or using appropriate self-help or coping strategies. A minority of responses suggested unintended outcomes, for instance, the person avoided the discussion (n = 4, 3.2%), no changes were observed (n = 4, 3.2%), or the person rejected the offer of help (n = 3, 2.4%). For the question “What did the person do as a result of your help?”, 70 responses were recorded before the course and 26 responses were recorded at six-month follow-up. A total of 8 responses were unclear or did not answer the question, resulting in 88 valid responses across time points and 132 codable actions. The most frequently reported actions taken by the person as a result of the help provided were that they sought professional help or support from others, talked or opened up more, or ceased or reduced self-harming behaviour. A small proportion of responses suggested that the person did nothing or there were no changes in their behaviour (n = 4, 3.0%) or that the person resumed self-harming (n = 2, 1.5%).

Before the course, 14 participants provided free-text responses that described their reasons for not helping a person engaging in NSSI and 8 participants provided reasons for not helping at follow-up. In total, 3 responses were unclear or did not answer the question, leaving 19 valid responses across both time points that contained 22 codable reasons. A total of 7 reasons (31.8%) related to the first aider lacking experience, knowledge, or confidence in supporting a person engaging in NSSI, 5 (22.7%) stated that other people were already supporting or were better placed to support the person, and 3 (13.6%) involved a lack of contact with the person. Other reasons included lacking opportunities to help (n = 2, 9.1%), choosing to focus on the person’s other mental health problems (n = 2, 9.1%), and not knowing the person well (n = 1, 4.6%), not having a good relationship with them (n = 1, 4.6%), or not wanting to embarrass them (n = 1, 4.6%).

### 3.10. Course Satisfaction and Course Feedback

Overall, participant feedback was very positive and all elements of the course were rated very highly (see Table 9). The average score was 5 (out of 5) for all course satisfaction measures, except for the measure which rated the novelty of course information (4 out of 5, SD = 1.0). Qualitative feedback indicated that group discussions were most helpful for assisting learning and the videos presented were helpful for applying the course content to “real life” contexts. Participants also found that the content was relevant, informative, and helped them to develop useful skills that could be applied across a range of contexts.

Areas where the course could be improved included allowing more time to cover course content, better alignment of the different learning materials (i.e., PowerPoint and course handbook), and the use of roleplays that could give participants the opportunity to practice their new skills and knowledge. The most common answers (i.e., those that received endorsement from at least 10% of the sample) are summarised in Table 10.

## 4. Discussion

The present uncontrolled trial used a pre-course, post-course, and six-month follow-up survey to evaluate the effects of Mental Health First Aid Australia’s *Conversations about Non-Suicidal Self-Injury* course on knowledge, stigma, social distance, and confidence in and quality of both intended and actual helping behaviours. Perceived acceptability of the course was also assessed using course satisfaction ratings and participant feedback. Participation in the course was associated with a significant increase in knowledge about NSSI, quality of intended helping behaviours, and confidence in enacting these behaviours. Participants who provided actual help to a person engaging in NSSI after the course reported significantly improved confidence in providing this support and high concordance with non-recommended helping behaviours. Additionally, both stigma and social distance reduced significantly over time. These findings are comparable with the outcomes of prior Mental Health First Aid course evaluations [19,20,21,32].

Participant knowledge about NSSI was moderate before the course and showed sustained improvement over time, with mastery being achieved by over three quarters of participants at post-course and follow-up. This was reflected in the participant satisfaction data where participants rated the content as mostly new (see Table 9). In clinical settings, high levels of knowledge about NSSI have been associated with a positive attitude towards and confidence in helping a person who had engaged in NSSI [33]. Conversely, parents of children who had engaged in NSSI reported that lack of knowledge contributed to feelings of shock and fear in response to disclosure from their child [34]. In a systematic review of NSSI disclosure responses, those who had disclosed their NSSI behaviour to others reported that negative, trivialising, or stigmatising responses were a deterrent to further help-seeking. Furthermore, feeling misunderstood when disclosing NSSI was associated with continuation of the behaviour [11]. The results of this evaluation suggest that the *Conversations about Non-Suicidal Self-Injury* course may promote appropriate responses to NSSI disclosures in the general population and provide participants with the information and skills needed to encourage further help-seeking.

Similar sustained improvements were demonstrated for confidence in intended help and confidence in actual help provided to a person who had engaged in NSSI. Confidence in intended help showed greater mean improvements from pre-course to six-month follow-up than confidence in actual helping situations. This discrepancy may be influenced by additional factors, such as the participant’s relationship to the person they are helping, which are present in real-world interactions and difficult to simulate in roleplay scenarios as taught in the course [35,36,37]. It may also be due to the fact that, upon reflection, participants may have been more confident in their anticipated helping behaviours in response to the Alicia vignette than compared to their actual helping behaviours. Nonetheless, further improvements in these outcomes may be achieved by increasing the duration of the *Conversations about Non-Suicidal Self-Injury* course and including more opportunities for interactive roleplays to practice their skills and increase confidence. These recommended changes are consistent with participant feedback requesting more time for skills practice within the course.

The quality of intended helping behaviours significantly increased at post-course and six-month follow-up. Concordance with non-recommended actions showed the greatest improvement, with participants’ reducing the number of non-recommended actions endorsed at the post-course assessment. However, mastery for intended recommended actions was higher at all time points than mastery for non-recommended actions, which may suggest that appropriate actions to take when supporting someone engaging in NSSI are clearer or more memorable than actions to avoid. Furthermore, some non-recommended actions (for example, telling them to stop) may reflect intuitive responses which could be related to participants’ lack of knowledge or confidence. Future iterations of the *Conversations about Non-Suicidal Self-Injury* course should consider these findings and aim to further reduce the number of non-recommended actions endorsed by members of the public. This could be achieved by incorporating more roleplaying activities in the course, which could help to clarify actions that may appear intuitive or helpful but should instead be avoided (for instance, telling the person how distressing their injuries are to you).

Interestingly, of the participants who reported that they had provided support to someone who engaged in NSSI after undertaking the course, the proportions for mastery were higher for non-recommended actions than recommended actions. These results may indicate that, although participants knew what actions they should take, there may be barriers present in real-life situations that are not measured in this study that prevent them from taking appropriate action, for example, the participant’s relationship to the person they are supporting or the context in which NSSI was disclosed or discovered [34,37,38]. Given that participants reported that the person they helped was most commonly a client/patient, it is possible that the actions listed in the scale may not have been appropriate for the context and affected mastery scores. Similar findings were present in evaluations of Mental Health First Aid Australia’s *Conversations about Suicide* course [19] and *Conversations about Gambling* course [20]. This finding may have been influenced by the low response rate at six-month follow-up and the lower response rate of those who reported helping someone who engaged in NSSI. Further investigation is needed to determine the cause of the disparity between mastery of recommended and non-recommended actions, which could be achieved by powering future studies to detect behavioural outcomes.

There are several possible explanations for why the non-recommended item “Ask Alicia about why she is injuring herself” was consistently incorrectly endorsed as both an intended and actual behaviour. First, the action may not have been appropriately distinguished from a similarly worded recommended action included in the guidelines that informed the course: “Ask the person questions about their self-injury, but avoid pressuring them to talk about it” [18]. This recommended action encourages gentle questioning about the self-injury, with the exact nature of the questions being left to the first-aider’s judgement, whereas the non-recommended action directs participants to ask about the reasoning behind the NSSI behaviour. It is possible that the non-recommended action statement was being misinterpreted as a result of the similarity in wording. Secondly, the vignette used to assess quality of intended helping behaviour may not have provided enough context for participants to distinguish whether the statement in the behavioural scale is a recommended or non-recommended action based on the course learning outcomes. This issue could be addressed by asking participants to elaborate on or explain why they chose each action. Thirdly, as people who engage in NSSI are up to five times more likely to experience suicidal ideation [7], identifying whether someone is suicidal using direct questioning is a component of the *Conversations about Non-Suicidal Self-Injury* course curriculum. The direct questioning style presented in the above non-recommended action may have been conflated with the recommended direct questioning to discern if a person is suicidal, rather than asking about why they are injuring themselves. A review of the course curriculum and delivery that incorporates instructor feedback is needed to establish whether this item is appropriate and effective in relation to the desired learning outcomes.

### Strengths, Limitations, and Future Directions

The *Conversations about Non-Suicidal Self-Injury* course was evaluated to determine the effectiveness of the course and it showed sustained improvement over time for nearly all outcomes. The sustained increase in knowledge over time may support reductions in misconceptions and stigma around NSSI. The course also significantly reduced stigma over time, an important finding given that participants in this study did not show highly stigmatising attitudes at pre-course assessment. Future research could assess whether the course reduces stigma in groups that are in regular contact with people who engage in NSSI and demonstrate high levels of stigma towards the behaviour. For example, emergency department nurses that had greater knowledge of NSSI exhibited lower levels of stigma [33]. Another strength of this evaluation was the high response rate, with 96.1% of course attendees consenting to participate, minimising the possible impact of selection bias.

The limitations of the evaluation include low response rates at follow-up and limited data about actual first aid helping actions provided after the course. Additionally, the absence of a control group makes it difficult to discern whether observed effects resulted from the course itself or extraneous factors that were not captured in this evaluation. The largely female, well-educated, English-speaking sample is consistent with participant groups from previous MHFA course evaluations [20,39] but limits generalisability outside of this demographic. Relatedly, the vignette of a young female used in this survey, while consistent with national demographic data on the prevalence of NSSI in Australia [3], may not be relatable to all participants, such as older people and males. This may have implications for how participants identify with, interpret, and respond to questions about how to assist the person in the vignette. Future research could include more diverse vignette characters. Furthermore, the course was developed for an Australian context, with panellists from the original Delphi study being drawn only from high-income countries. This means that, although the course is informed by a multi-faceted evidence base, it is unclear whether the curriculum would be relevant in middle- or low-income countries with different cultural profiles and levels of mental health literacy. Future research may seek to investigate the course efficacy across a longer time period, with a more diverse range of contexts and demographics.

The results of this study have several implications. First, this evaluation finds that the *Conversations about Non-Suicidal Self-Injury* course is safe, acceptable, and effective in achieving the aims of the course curriculum, adding to the evidence base for similarly evaluated early intervention programs developed by Mental Health First Aid Australia [19,20]. These results give confidence to licensed MHFA instructors that the course they are delivering is highly regarded by participants and has the potential to make a significant contribution to upskilling Australian communities. As these participants learn how to recognise, understand, and facilitate help-seeking for those experiencing NSSI, it may reduce the harms associated with NSSI [8].

The findings related to effectiveness and safety may also increase the likelihood that other licensees of MHFA courses, such as the USA, Canada, the UK, and those in other high-income countries, will adopt the *Conversations about Non-Suicidal Self-Injury* course. The results from the evaluation may also enable the effective promotion of the course to other Australian mental health community-based organisations, such as Lifeline and Mission Australia, who may include the course as part of their professional development training.

## 5. Conclusions

The present evaluation provides initial evidence to support the effectiveness and acceptability of Mental Health First Aid Australia’s *Conversations about Non-Suicidal Self-Injury* course. Consistent with previous evaluations of MHFA training, participants demonstrated improved knowledge of NSSI, reduced social distance and stigmatising attitudes, and increased confidence and quality of helping behaviours immediately after the course, with outcomes sustained over a 6-month period. Participant feedback indicated a high degree of satisfaction with course content and delivery, while also reporting areas for improvement that may inform future revisions of the course. Further research is needed to assess whether these outcomes are consistent across more diverse participant groups, how long the effects of the training last for, and whether actual helping actions provided are perceived as beneficial by the recipient of the support.

## Figures and Tables

**Table 1 ijerph-20-03749-t001:** Participant characteristics pre-course (n = 147).

Variable		
Age—M (SD)		45.8 (13.2)
Gender—N (%)		
	Male	31 (21.1)
	Female	114 (77.5)
	Other	2 (1.4)
Education—N (%)		
	Year 9 or lower	5 (3.4)
	Year 10, 11, or 12	24 (16.3)
	Certificate, Trade, or Apprenticeship	45 (30.6)
	University	63 (42.9)
	Other	10 (6.8)
Aboriginal or Torres Strait Islander—N (%)		1 (0.7)
Language other than English—N (%)		10 (6.8)
Postcode—N (%)		
	Urban	55 (37.4)
	Regional/Rural/Remote	87 (59.2)
	Missing	5 (3.4)
Previous training about NSSI—N (%)		19 (12.9)
Previous training in mental health—N (%)		102 (69.4)
Experience with NSSI—N (%)		
	Clients/patients	52 (35.4)
	Colleague	10 (6.8)
	Myself	11 (7.5)
	Friends	33 (22.5)
	Family	43 (29.4)
	Student	20 (13.6)
	Acquaintance	22 (15)
	No experience	37 (25.2)
	Rather not say	1 (0.7)
Reason for attending the		
*Conversations about Non-Suicidal Self-Injury* course—N (%)		
	Part of continuing education for workplace/profession	104 (70.8)
	Part of training for a volunteer job	48 (32.7)
	To support someone who engages in NSSI	40 (27.2)
	Past contact with someone who engages in NSSI	50 (34)
	They engage in NSSI	10 (6.8)
	Other	11 (7.5)

**Table 2 ijerph-20-03749-t002:** Observed means and standard deviations for outcome measures.

	Pre-Course		Post-Course		Follow-Up	
	(n = 147) ^#^		(n = 137) ^#^		(n = 72) ^#^	
	M	SD	M	SD	M	SD
Knowledge about NSSI	69.6	17.9	88.48	8.95	84.85	9.62
Social distance	1.9	0.7	1.66	0.71	1.63	0.57
Stigma	1.67	0.55	1.41	0.41	1.46	0.45
Confidence in helping vignette	2.89	0.89	4.03	0.67	3.9	0.75
Intended help—recommended actions, number concordant ^a^	7.19	1.26	7.66	0.94	7.54	1
Intended help—non-recommended actions, number concordant ^b^	6.89	1.92	7.93	1.49	7.86	1.44
Confidence in helping person with engaging in NSSI	2.98	0.98	-	-	3.8	0.81
Help provided—recommended actions, number concordant ^a^	5.27	2.42	-	-	5.48	2.21
Help provided—non-recommended actions, number concordant ^c^	9.51	0.83	-	-	9.83	0.47

Note: ^#^ number of participants who provided any data at each time point. ^a^ Out of a total of 8 recommended actions. ^b^ Out of a total of 9 non-recommended actions. ^c^ Out of a total of 10 non-recommended actions.

**Table 3 ijerph-20-03749-t003:** Mean changes and odds ratio (OR) over time from pre-course to post-course and pre-course to follow-up.

	Mean Change Over Time Contrasts									
	Pre-Course to Post-Course					Pre-Course to Follow-Up				
	M	95% CI	*p*	d	95% CI	M	95% CI	*p*	d	95% CI
Knowledge about NSSI ^b^	19.48	16.89, 22.07	**<0.001**	1.32	1.06, 1.58	15.09	11.73, 18.44	**<0.001**	0.97	0.66, 1.27
Social distance ^a^	−0.22	−0.30, −0.14	**<0.001**	−0.33	−0.57, −0.09	−0.22	−0.33, −0.11	**<0.001**	−0.41	−0.70, −0.11
Confidence in helping vignette ^a^	1.15	1.02, 1.29	**<0.001**	1.41	1.14, 1.68	1.02	0.85, 1.18	**<0.001**	1.19	0.88, 1.49
Intended help—recommended actions, number concordant ^b^	0.46	0.26, 0.70	**<0.001**	0.42	0.18, 0.65	0.31	0.001, 0.62	**0.016**	0.29	−0.01, 0.58
Intended help—non-recommended actions, number concordant ^b^	1.07	0.79, 1.34	**<0.001**	0.6	0.36, 0.84	0.79	0.43, 1.15	**<0.001**	0.54	0.25, 0.84
Confidence in helping person engaging in NSSI ^a^	-	-	-	-	-	0.79	0.44, 1.13	**<0.001**	0.87	0.44, 1.31
Help provided—recommended actions, number concordant ^b^	-	-	-	-	-	0.12	−0.60, 0.83	0.752	0.09	−0.33, 0.51
Help provided—non-recommended actions, number concordant ^b^	-	-	-	-	-	0.29	0.07, 0.51	**0.009**	0.42	−0.01, 0.84
	**OR**		**95% CI**		** *p* **	**OR**		**95% CI**		** *p* **
Stigma about NSSI	0.3		0.14, 0.63		**0.002**	0.5		0.20, 1.25		0.138

Note: ^a^ bias-corrected parameters based on 2000 bootstrapped replications. ^b^ Value is from model using transformed data to meet model assumptions. Bolded text indicates statistical significance.

**Table 4 ijerph-20-03749-t004:** Number and percent of participants who intended to undertake recommended actions to support Alicia ^a^.

Recommended Actions	Pre-Course (n = 143) ^#^		Post-Course (n = 137) ^#^		Six-Month Follow-Up (n = 67) ^#^	
	n	%	n	%	n	%
Directly express your concerns to Alicia about her injuries	127	88.8	126	91.9	62	92.5
Ask Alicia about her feelings that have led her to injure herself	124	87.3	131	95.6	63	94
Ask Alicia if she is having thoughts of suicide	119	83.2	131	95.6	62	92.5
Help Alicia find ways to make her life more manageable	133	93	133	97.1	65	97
Tell Alicia she can call you when she is feeling like injuring herself	135	93	131	95.6	63	94
Tell Alicia that there are sources of help and support available	140	97.9	134	97.8	66	98.5
Offer to help Alicia to see mental health treatment	133	93	133	97.1	64	95.5
Ask Alicia if there are things she can do that will help her delay injuring herself	120	83.9	131	95.6	60	89.5
Mastery of intended recommended actions ^b^	120	83.9	130	94.9	62	92.5

Note: ^#^ number of participants who completed at least 80% of the questions in the scale. ^a^ Participant rated they were “Likely” or “Very likely” to undertake the action. ^b^ Participants rated they were “Likely or “Very likely” to undertake at least 7 recommended actions.

**Table 5 ijerph-20-03749-t005:** Number and percent of participants who intended to undertake actions that were not recommended to support Alicia ^a^.

Non-Recommended Actions	Pre-Course (n = 143) ^#^		Post-Course (n = 135) ^#^		Six-Month Follow-up (n = 67) ^#^	
	n	%	n	%	n	%
Wait and see if her problems go away	11	7.7	3	2.2	2	3
Wait and see if her problems get worse	8	5.6	2	1.5	1	1.5
Wait and see if Alicia says that she thinks she might have a problem.	15	10.49	5	3.7	0	0
Ignore Alicia’s injuries because she could be doing this to get attention	3	2.1	1	0.7	2	3
Ask Alicia about why she is injuring herself ^b^	105	73.4	119	88.1	52	77.6
Let Alicia know how distressing her injuries are to you	48	33.6	39	28.8	15	22.4
Tell Alicia to stop injuring herself	14	9.8	5	3.7	3	4.5
Tell Alicia that if she continues to injure herself she will have life-long scars	15	11.1	10	7.4	6	8.9
Tell Alicia that self-injuring is making things worse for her parents	5	3.5	4	2.9	0	0
Promise Alicia that if she stops injuring herself you will reward her	11	7.7	4	2.9	1	1.5
Mastery of intended non-recommended actions ^c^	70	48.9	101	74.8	46	68.7

Note: ^#^ number of participants who completed at least 80% of the questions in the scale. ^a^ Participant rated they were “Likely” or “Very likely” to undertake the action. ^b^ Removed from statistical analysis and mastery calculation. ^c^ Participants rated they were “Unlikely” or “Very unlikely” to undertake at least 7 non-recommended actions.

**Table 6 ijerph-20-03749-t006:** Number and percent of participants who undertook actions to support a person that were recommended by MHFA training.

Recommended Actions	Pre-Course (n = 85) ^#^		Six-Month Follow-Up (n = 29) ^#^	
	n	%	n	%
Directly expressed concerns to them about their injuries	54	63.5	18	62.1
Asked them about their feelings that have led them to injure themselves	64	75.3	23	79.3
Asked them if they were having thoughts of suicide	54	63.5	19	65.5
Help them find ways to make their life more manageable or reduce their distress	67	78.8	26	89.7
Told them they could call you when they were feeling like injuring themselves	49	57.6	13	44.8
Told them that there are sources of help and support available	62	58.3	22	75.9
Offered to help them seek mental health treatment	59	70.2	22	75.9
Ask them if there were things you could do that would help them delay injuring themselves	39	46.4	16	55.2
Mastery of actions concordant with training ^a^	36	42.4	13	44.8

Note: ^#^ number of participants who completed at least 80% of the questions in the scale. ^a^ Participants rated that they had undertaken at least 7 recommended actions.

**Table 7 ijerph-20-03749-t007:** Number and percent of participants who undertook actions to support a person that were not recommended by MHFA training.

Non-Recommended Actions	Pre-Course (n = 84) ^#^		Six-Month Follow-Up (n = 29) ^#^	
	n	%	n	%
Waited to see if their problems went away	0	0	0	0
Waited to see if their problems got worse	1	1.2	0	0
Waited to see if the person said that they thought they may have a problem	5	5.9	1	3.4
Ignored their injuries because you thought they could be doing it to get attention	0	0	0	0
Ask them about why they were injuring themselves ^a^	41	48.8	17	58.6
Let them know how distressing their injuries were to you	15	17.9	2	6.9
Told them to stop injuring themselves	4	4.8	0	0
Told them that if they continued they will have life-long scars	6	7.1	1	3.4
Told them that their self-injuring is difficult for the people around them	7	8.3	1	3.4
Promised them that that if they stopped injuring you would reward them	0	0	0	0
I did not do anything	2	2.4	0	0
Mastery of actions concordant with training ^b^	76	90.5	29	100

Note: ^#^ number of participants who completed at least 80% of the questions in the scale. ^a^ Removed from statistical analysis and mastery calculation. ^b^ Participants rated that they avoided undertaking at least 9 actions that were not recommended.

**Table 8 ijerph-20-03749-t008:** Results of content analysis for the questions “What were the effects on the person of what you did?” and “What did the person do as a result of your help?”

Effect Category	Example	Number of Coded Effects (% ^#^)
**What were the effects on the person of what you did?**		
Seemed relieved/calmer/more positive	“They were less anxious after I talked to them.”	18 (14.5)
Talked/opened up to the first aider	“Talking more openly about this issue and mental health in general.”	17 (13.7)
Sought or said they would seek professional help	“He sought professional help and is now in a much better position mentally.”	17 (13.7)
Felt supported and heard	“I think the person felt cared for and may have delayed further NSSI.”	15 (12.1)
Discussion or implementation of self-help or coping strategies	“They said they felt supported and were open to talking about strategies to manage stress and anxiety before self-injuring.”	13 (10.5)
Self-harmed less often	“They did not harm themselves again.”	8 (6.5)
Became aware that the first aider or other supports were available to them	“Felt supported and knew there were services available and people who could support them.”	8 (6.5)
Seemed appreciative of the first aider’s support	“They said they felt supported and listened to. Appreciated being open to discuss and offer support. Felt they weren’t being judged.”	7 (5.7)
**What did the person do as a result of your help?**		
Sought professional help	“Made contact with mental health support and is seeing a doctor.”	39 (29.6)
Talked/opened up more	“Engaged better on further contact. Spoke more easily about their feelings.”	12 (9.1)
Stopped or reduced self-harm	“Didn’t self-harm again after we talked and went back to their psychologist, remained physically safe and saw professional help.”	11 (8.3)
Sought help or support (without specifying from where)	“Asked for help.”	11 (8.3)
Used self-help strategies, including those that help to delay the urge to self-harm	“She began seeking professional help, writing a journal when she felt she needed to self-injure and has been engaging in methods of coping with these feelings.”	10 (7.6)
Positive changes to emotions or behaviours	“Seemed to settle down.”	8 (6.1)
Sought help from informal supports (e.g., friends or family)	“The student had several long conversations with their friends about how they were feeling and the support they could offer them when struggling.”	7 (5.3)

Note: only categories with more than 5% of responses coded to them are reported in this table. Responses were able to be coded into multiple categories. ^#^ Percentages were calculated as a proportion of the number of codable effects for each question (n = 124 for “What were the effects on the person of what you did?” and n = 132 for “What did the person do as a result of your help?”).

**Table 9 ijerph-20-03749-t009:** Mean scores for course satisfaction measures (n = 133).

**Course Satisfaction Measure**	**Possible Range**	**M (SD)**
How new was the information?	1 (not at all new) to 5 (mostly new)	4 (1.00)
How much did you understand?	1 (none of it) to 5 (most of it)	5 (0.34)
How well was the program presented?	1 (very poorly) to 5 (very well)	5 (0.51)
How relevant was the content?	1 (not very relevant) to 5 (very relevant)	5 (0.62)
**Please rate how much you liked the following parts of the program:**	1 (liked very much) to 5 (did not like very much)	
Handbook		5 (0.72)
PowerPoint		5 (0.69)
Videos		5 (0.70)
Activities		5 (0.81)

**Table 10 ijerph-20-03749-t010:** Summary of qualitative course satisfaction data and feedback.

**Theme**	**n (%)**	**Illustrative Quote**
**What was most helpful?** (**n = 115)**		
Videos	23 (20.0)	“The videos were very good as they posed real life situations for the group to discuss.”
Group discussions	23 (20.0)	“Good group discussions which helped me process what we were learning”
Course content	22 (19.0)	“The relevance of the information and the strategies in dealing with NSSI.”
Real life examples	19 (16.5)	“There was opportunity for discussion and we looked at actual case studies as examples.”
Learning new skills	17 (14.8)	“The course was informative and gave useful tools and skills which can be applied both in my workplace and personal life.”
Group activities	14 (12.2)	“The course expanded on my basic knowledge. The role playing/scenarios were very helpful in this.”
**What could be improved? (n = 41)**		
More time to cover content (course is too short)	10 (24.4)	“More time to go more in-depth; to play out more scenarios to work through situations.”
**Improvements to course:**	8 (19.5)	“The PP [PowerPoint] and the workbook could blend more cohesively. We often flicked back and forth to find information.”
Handbook	4	
Videos	2	
Scripts	1	
PowerPoint	1	
More role plays	5 (12.1)	“Perhaps we could have a role play to use our new skills...”
Application of content	4 (9.8)	“I would appreciate a session directed more specifically to primary school/early teen. Although this was helpful.”

Note: only responses that received endorsement from at least 10% of the sample (n) population for each question are listed in this table.

## Data Availability

The data presented in this study are available in Appendix A CANSSI_data_Sep2022_article.

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
