# Peer review of "Evaluation of the Conversations about Non-Suicidal Self-Injury Mental Health First Aid Course: Effects on Knowledge, Stigmatising Attitudes, Confidence and Helping Behaviour"

_ijerph, 2023, doi:10.3390/ijerph20043749_

Round 1
Reviewer 1 Report
Thank you for the opportunity to review this manuscript which presents findings from an evaluation of an educational mental health first aid course on knowledge, stigma, confidence and helping related to non-suicidal self-injury. This study continues the fine tradition of published evaluation of MHFA and related content across the globe, continuing to build the evidence base for mental health literacy in mental health promotion, prevention and early intervention.
Introduction
1. line 33 - deliberately inflicted feels value-heavy and judgemental/stigmatising. Is there another way that this can be framed?
2. line 47 - can you provide an example of the particular types of thoughts/feelings?
3. line 74 - Given the international readership I suggest a footnote about MHFA and a url for more information.
4. line 77- might be worthwhile providing a couple of examples for those not familiar with the use of mental health problem and crises (as opposed to illness language etc).
5. line 82 - The phrase lived experience experts may not be familiar terminology to an international audience -suggest definition in parentheses or footnote.
6. line 78 – can you provide one or two examples of the specific demographic groups?
7. A sentence could be added about who will benefit, from reading this paper, and how?
Methods
1. line 109 - I think it would be valuable to reference that the study follows a similar process/uses measures described/reported in other published MHFA evaluation.
2. line 101 - For a reader who isn't familiar with the MHFA model, a brief note about accredited instructors would be valuable-what do they have to do to become accredited, and accredited by whom?
3. line 114 - A reference for survey monkey is desirable along with a descriptor, e.g., is online survey software or similar.
4. line 120 - Can you describe what the demographics/other questions included and whether these were standardised from previous eval/studies.
5. line 121 - in the demographics table, reason/motivation for participation is presented, but it isn't described here as a measure-could be included.
6. line 141 - Could you clarify the scoring for social distance-a 5 point scale, but scoring out of 7? is there a score overall for each scale? (opposed to the subscales)? I may have missed something/misunderstood.
7. line 143 – should this note Yap et al rather than just Griffith?
8. line 146 - It would be helpful to note that the confidence q has been used in other research with a corresponding reference.
Results
1. line 333 - At pre-course, 57.8% of participants (n = 85) stated they knew someone in their life 334 who had engaged in NSSI. At follow-up 19.7% of participants (n = 29) reported this.
This is interesting but I couldn’t see this in the discussion was this because their understanding of NSSI has improved and so they had different ideas about what was/wasn’t NSSI or some other reason?
Discussion/conclusion/future directions.
1. I would like to see the discussion and conclusion more clearly identify the new insights and implications of this research (theoretical, methodological, empirical, and practical/policy). Who are the actionable insights for (in MHFA but also beyond)? What might the findings mean for first aiders or for instructors or the community at large?
2. line 468-ish - some refs to broader literature would be helpful here regarding the influencing factors.
3. Might there be some value in talking to instructors? in terms of the influence of the delivery one some of the non-recommended actions?
4. A general comment - given the plethora of evaluation studies on MHFA, and the limitations that you rightly cite are relatively common across these studies, it would be worthwhile acknowledging that we do little to mitigate/ameliorate them and some practical methodological suggestions, or acknowledging that there are pragmatic considerations of evaluation from the field (RCTs not always possible or useful) and that realist evaluation is also important - e.g. - what works, for whom, in what respects, to what extent, in what contexts, and how?
5. While it very well may be required, finishing with more research is needed feels a little gratuitous! It would be helpful to hear a little more about some implications for policy/practice. How do contributions help researchers, policy-makers/practitioners do their jobs better?
General comments on style, expression
1. There are some dense/long sentences, and less active language than is desirable. Very long sentences – e.g., lines 81 - 85 etc…could be improved for readability.
2. Check the use of stigmatising in title and stigmatizing in text?
Reviewer 2 Report
Thank you for the opportunity to review the manuscript Evaluations of the Conversations About NSSI MHF Course, which was a pleasure to read. I commend the authors on a well-written and presented manuscript that reports on an important topic for international readers interested in improving the lives of individuals with lived experience of NSSI.
I have left comments on the manuscript itself. While there are several requests for more information/language change, most are suggestions/considerations. I look forward to reading the revised manuscript!
